# The Antibacterial and Antifungal Capacity of Eight Commercially Available Types of Mouthwash against Oral Microorganisms: An In Vitro Study

**DOI:** 10.3390/antibiotics12040675

**Published:** 2023-03-30

**Authors:** Silvia Di Lodovico, Tatiane Cristina Dotta, Luigina Cellini, Giovanna Iezzi, Simonetta D’Ercole, Morena Petrini

**Affiliations:** 1Department of Pharmacy, University of Chieti-Pescara, 66100 Chieti, Italy; 2Department of Dental Materials and Prosthodontics, School of Dentistry of Ribeirão Preto, University of São Paulo, São Paulo 14040-904, Brazil; 3Department of Medical, Oral and Biotechnological Sciences, University of Chieti-Pescara, 66100 Chieti, Italy

**Keywords:** oral microorganisms, chlorhexidine digluconate, cetylpyridinium chloride, mouthwashes, antimicrobial action

## Abstract

This work aimed to evaluate and compare the antimicrobial actions and effects over time of eight types of mouthwash, based on the impact of chlorhexidine on the main microorganisms that are responsible for oral diseases: *Enterococcus faecalis*, *Pseudomonas aeruginosa,* and *Candida albicans*. The mouthwashes’ antimicrobial action was determined in terms of their minimum inhibitory concentration (MIC), minimum bactericidal/fungicidal concentration (MBC/MFC), and time-kill curves at different contact times (10 s, 30 s, 60 s, 5 min, 15 min, 30 min, and 60 min), against selected oral microorganisms. All the mouthwashes showed a notable effect against *C. albicans* (MICs ranging from 0.02% to 0.09%), and higher MIC values were recorded with *P. aeruginosa* (1.56% to >50%). In general, the mouthwashes showed similar antimicrobial effects at reduced contact times (10, 30, and 60 s) against all the tested microorganisms, except with *P. aeruginosa*, for which the most significant effect was observed with a long time (15, 30, and 60 min). The results demonstrate significant differences in the antimicrobial actions of the tested mouthwashes, although all contained chlorhexidine and most of them also contained cetylpyridinium chloride. The relevant antimicrobial effects of all the tested mouthwashes, and those with the best higher antimicrobial action, were recorded by A—GUM^®^ PAROEX^®^A and B—GUM^®^ PAROEX^®^, considering their effects against the resistant microorganisms and their MIC values.

## 1. Introduction

Mouthwashes are currently used as adjunctive aids to the oral hygiene procedures (OH) of dental and interdental brushing, to prevent/reduce gingivitis [1]. A recent review has shown that the adjunctive use of antimicrobial mouthwashes for OH reduces plaque accumulation and gingivitis occurrence by 30% and 20%, respectively [2].

Chlorhexidine gluconate (1,1′-hexamethylene bi [5-(p-chlorophenyl) biguanide] di-D-gluconate) (CHX) (Figure 1) is widely employed for clinical use, especially in dentistry as an antiseptic mouthwash [1]. Developed in the 1940s and initially marketed as a general disinfectant, it only became available as a mouthwash in 1976 [3]. It is a broad-spectrum anti-microbial agent that disrupts cellular membranes, with a remarkable action against oral pathogenic microorganisms and effect on biofilm formation and plaque accumulation [4,5]. Chlorhexidine’s mechanism of action begins through the attraction of a cationic molecule to the surface of a negatively charged bacterial cell, causing a bond to form between both [3]. Through passive diffusion, a penetration occurs through the bacterial cell wall, damaging it and compromising its integrity, causing the precipitation and coagulation of the entire cytoplasmic content [6].

Cetylpyridinium chloride (1-hexadecylpyridinium chloride) (CPC) (Figure 2), a quaternary ammonium compound, is also frequently used in dental practices within different mouthwashes and dentifrices. Its antimicrobial action is correlated to the hydrophobicity of the side chain, and with its capability to damage the cell membrane, it affects the growth of Gram-positive and Gram-negative bacteria. It also showed a remarkable anti-biofilm action against the methicillin-resistant *Staphylococcus aureus*, *Streptococcus mutans,* and *Veillonella parvula* biofilm formations [7].

Oral cavity homeostasis is affected by microorganisms and their products. Different microbial species are responsible for causing dental caries, plaque formation, gingivitis, periodontitis, and oral soft tissue diseases. Therefore, the use of mouthwashes could represent an aid for preventing these oral diseases [8,9,10].

In particular, *Enterococcus faecalis*, *Pseudomonas aeruginosa*, and *Candida albicans* are microorganisms that are usually found in the oral cavity, isolated from endodontic infections, periodontal diseases, denture stomatitis, and other complications and oral diseases [11]. *E. faecalis*, a Gram-positive bacterium, is responsible for endodontic diseases and is one of the species that is most involved in refractory pathologies [12]. It is capable of producing biofilm, invading dentinal tubules, and developing a mono-infection, in addition to being able to resist endodontic disinfectants and irrigants, and having the ability to survive for long periods [13,14]. *P. aeruginosa*, a Gram-negative bacterium, has also been isolated in the root canal, with a high prevalence in the oral cavities of hospitalized patients. This bacterium has a high virulence, and it is responsible for persistent infection due to its antibiotic resistance and remarkable ability to form a biofilm [15]. *P. aeruginosa* is motile and, consequently, it has shown a greater ability to leak the implant/abutment interface in comparison to other bacterial species [13,14]

*C. albicans*, a yeast, is a commensal oral microorganism that does not cause problems in healthy individuals, but can switch to an opportunistic pathogen. It can produce different diseases, such as denture stomatitis, and is involved in the failure of dental implant devices [16]. Due to its hydrophobicity on the cell surface, it has a remarkable ability to adhere to inert surfaces, such as dental restorations, acrylic prosthesis bases, and orthodontic metal appliances, thus causing a predisposition for oral infections [17,18]. According to Ardizzoni et al., 2018 [14], in addition to its ability to adhere to inert materials, *C. albicans* can also adhere to the biotic surfaces within the oral cavity, such as the mucosal epithelia and the surface of the teeth, through other processes, such as the interactions between the *C. albicans* adhesins and epithelial receptors.

According to the literature, patients with gingivitis who practice mechanical oral hygiene and use chlorhexidine mouthwash experience significantly lower plaque and gingivitis scores [5,14]. Additionally, a meta-analysis by Costa et al. [19] revealed that using mouthwashes that contained a chlorhexidine base reduced the probing depth in patients with chronic periodontitis. This is in addition to several other studies, in which chlorhexidine gluconate was shown to be effective in reducing the virulence of bacteria and yeasts such as *E. faecalis* and *C. albicans* [14,20,21]. The long-term use of antiseptics may produce a resistance in vivo, due to the exposure to sublethal concentrations that has arisen over the last century [22]. Antimicrobial resistance is a global challenge, and in particular, an increase in antimicrobial-resistant microorganisms in the oral cavity is the main cause of the failure of traditional treatments. The oral cavity is a ready environment for horizontal gene transfer because of the close proximity of the bacteria in plaque and the availability of exogenous DNA that passes through the oral cavity [23]. For this reason, it is important to control and prevent the infection that is associated with resistant microorganisms.

The market is widely supplied with various types of mouthwash, which contain very similar or equal percentages of chlorhexidine and cetylpyridinium. However, is it sure that the same concentrations of the active principles will determine the same antibacterial and antifungal activity? The hypothesis of this study is that the additive compounds that are contained inside these mouthwashes could play a fundamental role in the efficacy of the products.

This work aimed to evaluate and compare the antimicrobial action of eight types of mouthwash at different contact times (10 s, 30 s, 60 s, 5 min, 15 min, 30 min, and 60 min), against reference and clinical strains of the main resistant microorganisms that are responsible for oral diseases: *Enterococcus faecalis* ATCC 29213, *E. faecalis* SDL1, *Pseudomonas aeruginosa* ATCC 15442, *P. aeruginosa* BM, *Candida albicans* ATCC 10231, and *C. albicans* S5. The reference strains that were used in this work were the main microorganism test that was used for the evaluation of the bactericidal and fungicidal effects of antiseptic medicinal products (European pharmacopeia 10.0 07/2015:50111). The final aim was to suggest whether these mouthwashes were able to prevent and control the microbial proliferation by tackling the antimicrobial resistance.

## 2. Results

### 2.1. Mouthwashes Evaluated

The mouthwashes that were used in the study are listed in Table 1.

### 2.2. Antibacterial/Antifungal Activity

The eight types of mouthwash that were used in this study, which are shown in Table 1, showed a remarkable antimicrobial action against the tested microorganisms, with MIC values ranging from 0.02% to 50%. In general, the best antimicrobial effect was obtained with A—GUM^®^ PAROEX^®^ 0.12% CHX + 0.05% CPC and B—GUM^®^ PAROEX^®^ 0.06% CHX + 0.05% CPC against all the tested microorganisms. All the mouthwashes displayed a remarkable effect against *C. albicans* ATCC 10231 and *C. albicans* S5, with MICs ranging from 0.02% to 0.09%. The MBCs/MFCs ranged from 0.02% to 50%. The MBCs of *E. faecalis* ATCC 29212 and *E. faecalis* SDL1 were more than the MICs (with one/two/three/four steps more than the MICs). For *P. aeruginosa* ATCC 15442, the MBCs were one or two steps more than the MICs. The MFCs were similar to the MICs, except for G—Eludril Classic with *C. albicans* ATCC 10231 and D—CURASEPT ADS 205 for *C. albicans* S5. The G—Eludril Classic was ineffective against *P. aeruginosa* with an MIC and MBC of >50% (Table 2).

### 2.3. Antibacterial/Antifungal Activity over Time

The results that were obtained with the time-kill curves for *E. faecalis* ATCC 29212 showed a significant (*p* < 0.05) CFU/mL reduction, with respect to the inoculum. Except for the E—PERIOAID^®^ Intensive Care, a reduction trend at a short contact time (10 s) was registered. A significant (*p* < 0.05) CFU/mL reduction in the time was recorded with A—GUM^®^ PAROEX^®^ at all the tested times, and with B—GUM^®^ PAROEX^®^ at 10, 30, and 60 s. The C—CURASEPT ADS 212 CFUs/mL was statistically significant (*p* < 0.05) only at 5, 30, and 60 min with respect to the 10 s, and D—CURASEPT ADS 205 CFUs/mL was significant at 1, 5, and 30 min with respect to 30 s. E—PERIOAID^®^ Intensive Care and F—PERIOAID^®^ Active Control showed a significant (*p* < 0.05) CFUs/mL at 10, 30, and 60 s, and 30 s, 5, and 30 min, with respect to the other tested times, respectively. G—Eludril Classic and H—Eluday Care CFUs/mL were statistically significant at 10 and 30 s, respectively. The results demonstrated the good antibacterial and antifungal efficacy of all the mouthwashes at short contact times (*p* < 0.05), which was maintained with A—GUM^®^ PAROEX^®^ and B—GUM^®^ PAROEX^®^ at 60 min, with a statistical difference obtained for all the mouthwashes at the same tested time (Figure 3).

The time-kill curves that were obtained for *E. faecalis* SDL1 displayed a similar trend that was registered with all the tested mouthwashes. As shown in Figure 4, A—GUM^®^ PAROEX^®^ and B—GUM^®^ PAROEX^®^ showed significant (*p* < 0.05) CFUs/mL values at all the tested times; C—CURASEPT ADS 212 and D—CURASEPT ADS 205 CFUs/mL were significant (*p* < 0.05) at 30 s and at 10, 30 s, and 5 min, respectively; F—PERIOAID^®^ Active Control and G—Eludril Classic showed a significant CFUs/mL at 10, 30, and 60 s, and at 10 and 30 s, respectively. The best anti-bacterial effect was obtained with D—CURASEPT ADS 205 (at 30 s) (*p* < 0.05) for its MIC value, which was the highest MIC value. Considering the MICs, A—GUM^®^ PAROEX^®^’s MIC value, which was the lowest registered concentration, displayed the best CFU/mL reduction. The B—GUM^®^ PAROEX^®^’s MIC value (00.9%) lost its antimicrobial effect with a longer contact time (Figure 4).

Except for C—CURASEPT ADS 212, the time-kill curves that were obtained with *P. aeruginosa* ATCC 15442 displayed a general CFU/mL reduction (*p* < 0.05), with respect to the inoculum of all the tested mouthwashes. A higher antimicrobial effect was observed for a long time with all the tested mouthwashes. With A—GUM^®^ PAROEX^®^, a reduction of CFUs/mL, proportionally to the time, with <100 CFUs/mL from 15 min and a statistical difference (*p* < 0.05) were recorded at 10 and 30 s. With B—GUM^®^ PAROEX^®^, the best effect was recorded after 5 min with a statistical significance (*p* < 0.05) at 10 and 30 s. The other mouthwashes showed remarkable CFU/mL reductions after 10 s, reaching a significant (*p* < 0.05) reduction from 5 min. Only E—PERIOAID^®^ Intensive Care showed a statistical significance (*p* < 0.05) CFUs/mL at all the tested times. G—Eludril Classic was used at a final concentration of 50%, displaying a significant (*p* < 0.05) (with respect to the inoculum) antimicrobial action with <100 CFU/mL at all the tested times (Figure 5).

As shown in Figure 6, except for A—GUM^®^ PAROEX^®^, C—CURASEPT ADS 212, D—CURASEPT ADS 205, and E—PERIOAID^®^ Intensive Care, the mouthwashes showed a relevant antimicrobial action against *P. aeruginosa* BM immediately after 10 s of contact time. All the tested mouthwashes displayed a significant (*p* < 0.05) CFU/mL reduction in respect to the inoculum after 1 min. In particular, the significant (*p* < 0.05) A—GUM^®^ PAROEX^®^, B—GUM^®^ PAROEX^®^, and D—CURASEPT ADS 205 CFUs/mL were obtained at 10 and 30 s, in respect to the other contact times. A remarkable action has been recorded with B—GUM^®^ PAROEX^®^ and H—Eluday Care, obtaining 2.48 × 10^5^ ± 1.36 × 10^5^ and 2.42 × 10^5^ ± 2.04 × 10^5^ CFU/mL after 10 min, with this reduction being conserved in time and reaching <100 CFU/mL from 15 min. With C—CURASEPT ADS 212 and E—PERIOAID^®^ Intensive Care, a significant (*p* < 0.05) CFUs/mL was recorded at all the tested times, except at 15, 30, and 60 min. F—PERIOAID^®^ Active Control displayed a significant CFUs/mL only at 1 min. G—Eludril Classic was used at a final concentration of 50%, displaying a significant (*p* < 0.05) antimicrobial action with <100 CFU/mL at all the tested times. In comparing the effects of the mouthwashes, significant differences (*p* < 0.05) were obtained with: A—GUM^®^ PAROEX^®^ at 10, 30, and 60 s, in respect to the other mouthwashes; B—GUM^®^ PAROEX^®^ at 10, 30, and 60 s, with respect to C, D, and E; C—CURASEPT ADS 212 at all the tested times, in respect to the other mouthwashes; D—CURASEPT ADS 205 at 10, 30, and 60 s, in respect to the other mouthwashes; E—PERIOAID^®^ Intensive Care at 10, 30, 60 s, and 5 min, in respect to the other mouthwashes; F—PERIOAID^®^ Active Control at 10 and 30 s vs. A, C, an D at 60 s vs. all the mouthwashes; G—Eludril Classic at 10, 30, and 60 s vs. A, C, E; and H at 10, 30, and 60 s vs. A, C, D, and E (Figure 6).

Except for G—Eludril Classic and H—Eluday Care, the tested mouthwashes displayed a relevant effect against *C. albicans* ATCC 10231 at short times. A remarkable CFU/mL reduction was obtained with A—GUM^®^ PAROEX^®^ at 10 s, and it was then stabilized in time. The B—GUM^®^ PAROEX^®^ effect was remarkable at a short time, and then it seemed to lose its effect. C—CURASEPT ADS 212 and D—CURASEPT ADS 205 showed the same trend, with a relevant effect in terms of the CFU/mL reduction. E—PERIOAID^®^ Intensive Care displayed the same effect as A—GUM^®^ PAROEX^®^, with a major effect at short times. G—Eludril Classic displayed an antimicrobial effect only after 5 min of contact time, while H—Eluday Care showed a significant effect at 10 s (Figure 7).

The significant effect, in terms of the CFU/mL reduction, against *C. albicans* S5 was detected with all the tested mouthwashes at a short time, 10 s, without a statistical difference (*p* > 0.05). After this contact time, except for D—CURASEPT ADS 205 at 30 s, C at 1 min, and H—Eluday Care at 5 min, the *C. albicans* S5 CFUs/mL was similar to the CFUs/mL of the inoculum (*p* > 0.05). The best effect was registered with G—Eludril Classic (at 10 s) at an MIC value of 0.04%. Although D—CURASEPT ADS 205 was used at the highest MIC value, it appeared to have less effect than the other mouthwashes, especially at 30 s. Considering the MIC values, B—GUM^®^ PAROEX^®^ (0.02%) was the mouthwash that showed the most major CFU/mL reduction, with respect to the inoculum (Figure 8).

## 3. Discussion

In this in vitro study, the antimicrobial activity of commercially available mouthwashes, which were based on chlorhexidine, were evaluated and compared at different concentrations (0.12%, 0.10%, 0.06, and 0.05%) against the microorganisms of oral infections. A combination of mechanical brushing and chemical agents effectively dealt with such oral problems [6,19].

The MIC values ranged from 0.02% to 50%, and all the mouthwashes showed better effects against *C. albicans*, with values ranging from 0.02% to 0.09%. The use of chlorhexidine digluconate and its antimicrobial action against *C. albicans* have been reported and documented in previous studies [5,24,25]

According to Varoni et al. [26], lower doses of CHX (0.06%) provided a comparable impact on reducing biofilm formation and controlling gingivitis, when compared to formulations with higher CHX concentrations. However, in the study by Chavarría-Bolaños et al. [24], the authors observed that the 0.12% CHX mouthwash was superior to the lower concentrations of 0.06% and 0.03%. In this study, it was observed that the tested mouthwashes were relevant, with remarkable reductions of *C. albicans* ATCC 10231 and *C. albicans* S5 in short times, at both concentrations. This finding is especially significant when considering the populations that are sensitive to oral fungal infections, such as removable prosthetic users or immunocompromised individuals. A low-dose chlorhexidine mouthwash being used frequently by these patients may provide significant therapeutic advantages [24].

The results of this study are in agreement with the previous literature, especially for *E. faecalis* and *P. aeruginosa*, which showed notable increases in the MIC values for MBC, denoting the need for higher concentrations of mouthwash to achieve these bactericidal capabilities. In the MIC results, it was also noted that there is a need for higher concentrations of mouthwashes against *P. aeruginosa* when compared to *E. faecalis*. Furthermore, the type of action of the mouthwash was dose-dependent, in which lower doses of chlorhexidine have been reported to have bacteriostatic qualities, whereas higher amounts have been reported to have bactericidal capabilities [6,26]. These data are in accordance with some authors [3,27,28], which shows that the biocidal activity of CHX was more effective against Gram-positive bacteria due to the greater negative charge of the cell.

According to Van Strydonck et al. [5], rinsing with 10 mL of chlorhexidine-based mouthwash for 60 s, twice a day, could inhibit plaque growth by 60% and reduce gingivitis by 50–80%. Through the results of this study, it was possible to verify large reductions in bacterial colonization over short periods. Additionally, despite these data, relevant reductions were notorious for a longer contact time against *P. aeruginosa*. Furthermore, the obtained results suggested that higher doses of mouthwash were useful for the decontamination regimens for the decolonization of *P. aeruginosa* activity in the oral cavity, according to the MIC and MBC values.

Mouthwash G was ineffective against *P. aeruginosa*, which could be explained by its absence of cetylpyridinium chloride (CPC). CPC is frequently used and included in dental practice, through consumer products such as mouthwashes and toothpaste [7]. It is a quaternary pyridinium antiseptic that shows bactericidal activity against some Gram-positive bacteria, and, when in higher concentrations, against some Gram-negative bacteria. As a cationic drug, it binds to negatively charged bacterial cell membranes and penetrates them, causing a bacterial cell component leakage, a disruption of the bacterial metabolism, a reduction in cell growth, and finally, bacterial cell death. CPC also inhibits the growth and accumulation of bacterial biofilm, which can help to decrease and control gingivitis and plaque [29,30,31].

Becker and collaborators [32], in their study, evaluated the action of a mouthwash, which was based on chlorhexidine combined with cetylpyridinium chloride, in the reduction of the living cells in oral biofilms that were adhered to the surface of hydroxyapatite and treated titanium, and emphasized the antimicrobial potential of mouthwashes that contained 0.05% CHX and 0.05% CPC. According to the same author [32], CPC had a moderate efficacy, if used alone; however, when combined with CHX, a synergistic effect increased its antimicrobial activity. This also corroborates the study by Santos et al. [33], who found positive results with the combination of CHX and CPC in reducing plaque levels and bacterial counts. A double-blind, randomized controlled trial evaluated the adjunctive use of 0.05% CPC and 0.05% CHX vs. 0.2% CHX, and discovered that both of them improved the plaque and gingivitis indices [34]. Additionally, in the study of Escribano et al. [35], a mouthwash with a low concentration of CHX (0.05%), combined with 0.05% cetylpyridinium chloride, showed significant reductions in the microbial loads of saliva and in the gingival sulcus.

In comparison, mouthwashes A and B were the most effective in fighting bacteria, according to the time-kill curves. Indeed, chlorhexidine is effective against bacteria and fungi, and its use in different concentrations has been reported in previous studies, denoting its antimicrobial action [24,36]. The antimicrobial effect of chlorhexidine at high concentrations against *E. faecalis* has been widely reported in the literature. In this study, concentrations of 0.12% and 0.06% were more effective against *E. faecalis*. The study of such a microorganism in dentistry is of paramount importance, since this bacterium is a component of the normal oral microbiota, and is of great interest for endodontics, as it is one of the species that is most involved in endodontic failure [37], in addition to showing a resistance against root canal disinfectants [24].

The obtained results stimulate new in vivo studies for analysing the shared interaction between the immune system and microorganisms, considering the existence of multispecies biofilms in periodontal and dental tissues [38]. However, although these results are in vitro, all the bacterial species that were used in this experiment are commonly found in the oral cavity, which makes our results relevant. Effectively, the use of ATCC strains may behave differently from the planktonic bacteria that was obtained from different populations (25). Therefore, in our study, we also used strains that were acquired directly from human patients, such as *Enterococcus faecalis* SLD1, *Pseudomonas aeruginosa* BM, and *Candida albicans* S5.

The concentrations of CHX of the mouthwashes that were used should be considered, since, at high concentrations, there is a possibility of adverse side effects, such as dental discoloration and stain, changes in taste, xerostomia, erosions, and ulcerations in the mucosa [3,4,8]. Another essential factor to consider is the development of bacterial resistance, which is considered to be a serious adverse effect [3,8]. Little is known about these risks involving the use of CHX. Therefore, health professionals should be encouraged to use chemical agents only when indicated, individualizing each patient.

## 4. Materials and Methods

### 4.1. Mouthwashes

The mouthwashes that were used in the study and their information are shown in Table 1. Mouthwashes A and B were purchased from GUM PAROEX (Saronno, Italy); C and D from CURASEPT S.p.A (Saronno, Italy); E and F from DENTAID, S.L. (Cerdanyola, Spain); and G and H from Pierre Fabre ORAL CARE (Lisboa, Portugal).

### 4.2. Substances

For the experiments, the interfering substance and the neutralizing agent were prepared. The interfering substance was obtained by mixing 0.3 g/L of bovine albumin in sterilized water, which was then filtered with filters at 0.45 mm. The neutralizing agent was used to neutralize the mouthwashes’ antimicrobial actions. It was prepared by mixing 3.0% Tween 80 with 0.3% L-α-lecithin and 0.5% sodium thiosulfate.

### 4.3. Microorganism Cultures

*Enterococcus faecalis* ATCC 29212, *E. faecalis* SDL1, *Pseudomonas aeruginosa* ATCC 15442, *P. aeruginosa* BM, *Candida albicans* ATCC 10231, and *C. albicans* S5 were used in the study. These clinical strains were isolated from the oral cavity of dental patients that underwent dental procedures at the Dental Clinic of the University of Chieti (reference number: BONEISTO N. 22-10.07.2021, University G. d’Annunzio Chieti-Pescara, 10 July 2021). The clinical strains were characterized by their susceptibility to the antibiotics/antifungals that are commonly used in therapy. The clinical strains were characterized by their antimicrobial susceptibility. *C. albicans* S5 was sensible to antifungal drugs that are commonly used in therapy. *E. faecalis* SDL1 was a resistant strain to Amoxicillin and Vancomycin. *P. aeruginosa* BM was a resistant strain to Amikacin, Amoxicillin, Aztreonam, Cefotaxime, Ceftazidime, Ceftriaxone, Cefalotin, Ciprofloxacin, Gentamicin, Levofloxacin, Nitrofurantoin, Piperacillin, and Tobramicin.

After this characterization, the microorganisms were stocked at −80 °C in the Departments of Pharmacy and of Medical, Oral, and Biotechnological Sciences of “G. d’Annunzio”, University of Chieti, Italy, until their use. For the experiments, the bacteria were cultured in a Trypticase Soy Broth (TSB, Oxoid, Mila, Italy) and incubated at 37 °C overnight in aerobic conditions, and then refreshed for 2 h at 37 °C in an orbital shaker in aerobic conditions. The cultures were standardized to an optical density of 600 nm (OD_600_) = 0.125, which corresponds to 10^7^ CFU/mL. The *Candida albicans* strains were cultured in RPMI 1640 + 2% glucose and standardized to OD_600_ = 0.15, which corresponds to 10^7^ CFU/mL.

### 4.4. Mouthwashes Antimicrobial Action

The antimicrobial actions of the mouthwashes were evaluated by their minimal inhibitory concentrations (MIC) and minimal bactericidal/fungicidal concentrations (MBC/MFC), which were obtained by the microdilution method, according to the Clinical and Laboratory Standards Institute [CLSI], 2018. For the test, the standardized bacterial suspensions (described above) were diluted 1:100 in a cation-adjusted Mueller–Hinton broth (CAMHB, Oxoid, Milan, Italy), in order to obtain a final inoculum of 5 × 10^5^ CFU/mL. The yeast suspensions were diluted 1:10 in RPMI + 2% glucose to obtain a final inoculum of 5 × 10^6^ CFU/mL. The broth cultures were used to inoculate 96-well microtiter plates, which had previously contained the serially diluted mouthwashes (2-fold dilutions). The mouthwashes were tested in the final concentrations, ranging from 50% to 0.04%. As a negative control, only medium (without the strains) was added to the different concentrations of the mouthwashes. The lowest concentration of the mouthwashes that was required to inhibit the microbial growth was defined as the MIC. The MBCs/MFCs were determined by sub-culturing 10 μL of the suspensions from the MICs on a Mueller Hinton agar (MHA, OXOID, Milan, Italy) for the bacteria, and on a Sabouraud Dextrose agar (SAB, OXOID, Milan, Italy) for *C. albicans*. The MBCs were the lowest concentrations of the mouthwashes that inhibited the microbial growth on the plates. Each determination was performed in three independent experiments, which were each duplicated [38].

### 4.5. Time Kill Curves

To define the effect of each mouthwash at different contact times against the tested microorganisms, time-kill curves (ASTM E2315-16) were performed. For the experiments, the standardized microorganisms that are described above were inoculated with the mouthwashes at the final concentrations of the MIC values, and at regular contact times, the CFUs/mL were determined. Briefly, 1 mL of the standardized microbial suspension and 1 mL of the interfering substance were mixed for 2 min, and then 8 mL of each mouthwash, at each final concentration of the MIC values, was added. At regular contact times (10 s, 30 s, 60 s, 5 min, 15 min, 30 min, and 60 min), 0.1 mL of the mix was put into the neutralizing agent, diluted, and the CFUs/mL was determined by spreading the mix on a Trypticase Soy agar (TSA, Oxoid, Milan, Italy) for the bacteria, and on SAB for the yeasts. The plates were incubated for 24–48 h in aerobic conditions at 37 °C (Figure 9).

### 4.6. Statistical Analysis

The statistical analysis was performed using SPSS Statistics for Windows, version 21 (IBMSPSS Inc., Chicago, IL, USA). A Levene’s test permitted the evaluation of the homogeneity of the variables. An analysis of variance (ANOVA) and Fisher’s least significant difference (LSD) test were performed to assess the presence of intragroup and intergroup differences that were statistically significant.

## 5. Conclusions

The results demonstrate significant differences in the antimicrobial actions of the tested mouthwashes, although all of them contained chlorhexidine and most of them also contained cetylpyridinium chloride. The highest antimicrobial action was recorded by A—GUM^®^ PAROEX^®^A and B—GUM^®^ PAROEX^®^, considering their effect against the resistant microorganisms and MIC values.

The obtained results suggest that, although the presence of CHX and CPC in the different concentrations provides an effective antimicrobial action, additional molecules in the mouthwash composition can significantly empower or diminish their effectiveness.

Regular and correct oral hygiene procedures, with the addition of suitable mouthwashes, in terms of their concentrations and efficacy against resistant microorganisms, can be considered to be a real solution for controlling and preventing infections of the oral cavity.

This is a preliminary in vitro study to compare the efficacy of different commercially available types of mouthwash against *E. faecalis*, *P. aeruginosa,* and *C. albicans* strains. Future work will be performed to confirm these results, with other oral aerobic and anaerobic microorganisms that are associated with oral diseases, and to overcome the study’s limitation of its lack of in vivo study aims for evaluating the CHX and CPC mouthwash effects on patients over time.

## Figures and Tables

**Figure 1 antibiotics-12-00675-f001:**
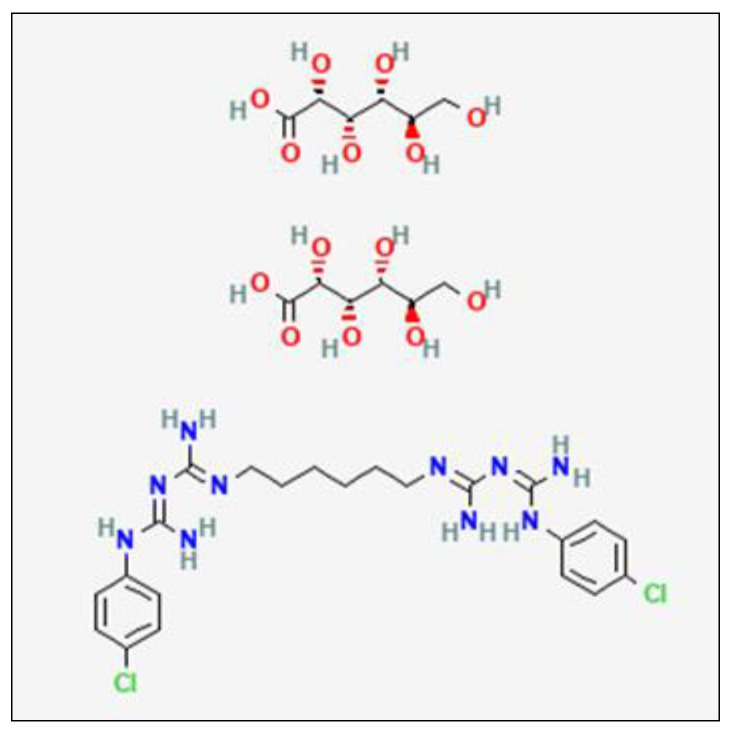
Chemical structure of chlorhexidine gluconate (CHX). The image has been downloaded by PubChem, https://pubchem.ncbi.nlm.nih.gov/compound/9552081. National Center for Biotechnology Information. “PubChem Compound Summary for CID 9552081, Chlorhexidine Gluconate” PubChem, https://pubchem.ncbi.nlm.nih.gov/compound/Chlorhexidine-Gluconate. Accessed on 1 February 2023.

**Figure 2 antibiotics-12-00675-f002:**
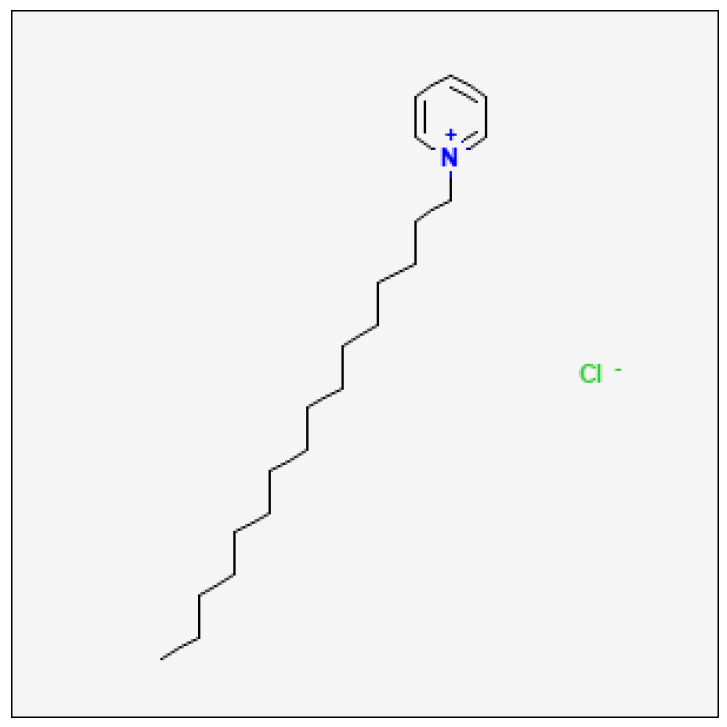
Chemical structure of cetylpyridinium chloride (CPC). The image has been downloaded by PubChem, https://pubchem.ncbi.nlm.nih.gov/compound/22324. National Center for Biotechnology Information. “PubChem Compound Summary for CID 22324, Cetylpyridinium chloride monohydrate” *PubChem*, https://pubchem.ncbi.nlm.nih.gov/compound/Cetylpyridinium-chloride-monohydrate. Accessed on 1 February 2023.

**Figure 3 antibiotics-12-00675-f003:**
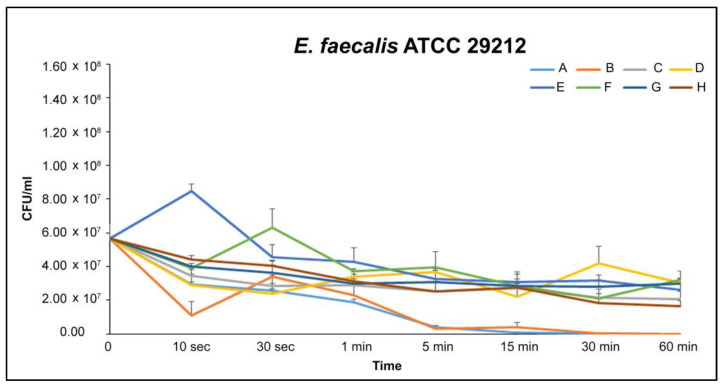
Time-kill curves with E. faecalis ATCC 29212 for the A-H mouthwashes. For the legend of the name of mouthwashes see Table 1.

**Figure 4 antibiotics-12-00675-f004:**
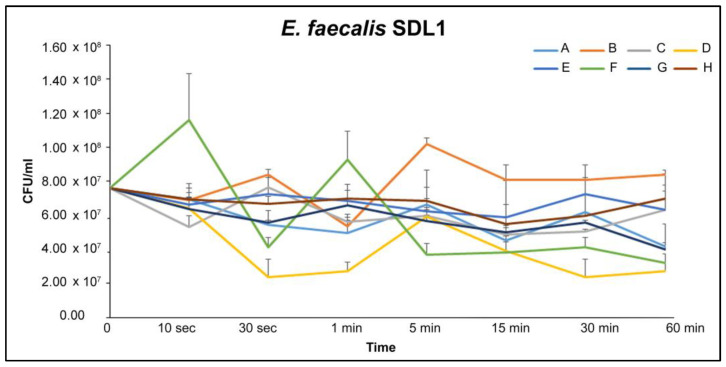
Time-kill curves with *E. faecalis* ATCC SDL1 for the A-H mouthwashes. For the legend of the name of mouthwashes see Table 1.

**Figure 5 antibiotics-12-00675-f005:**
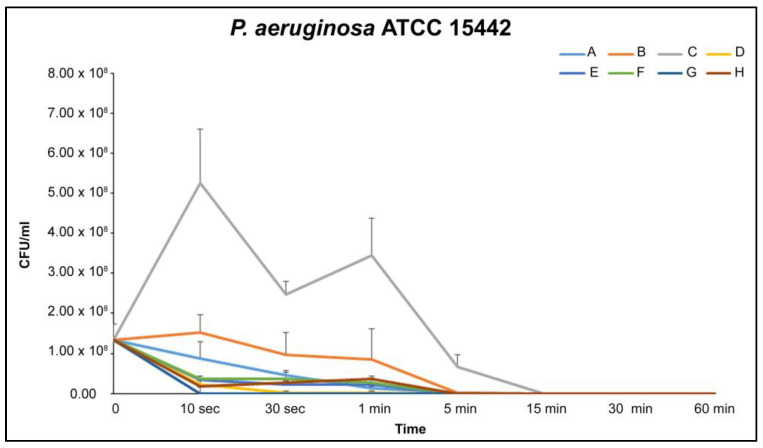
Time-kill curves with *P. aeruginosa* ATCC 15442 for the A-H mouthwashes. For the legend of the name of mouthwashes see Table 1.

**Figure 6 antibiotics-12-00675-f006:**
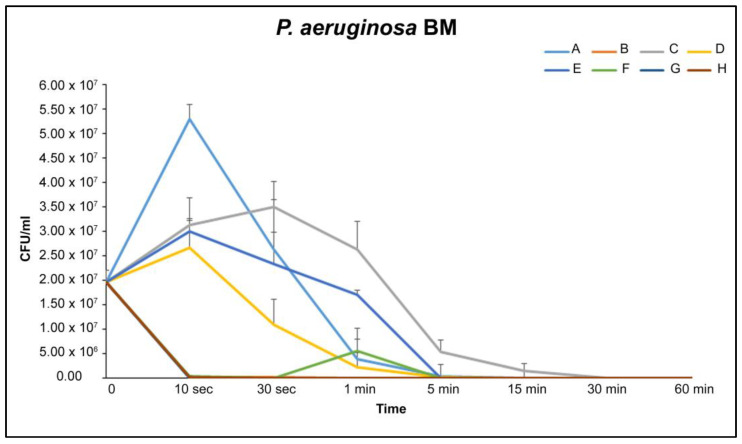
Time-kill curves with *P. aeruginosa* BM for the A-H mouthwashes. For the legend of the name of mouthwashes see Table 1.

**Figure 7 antibiotics-12-00675-f007:**
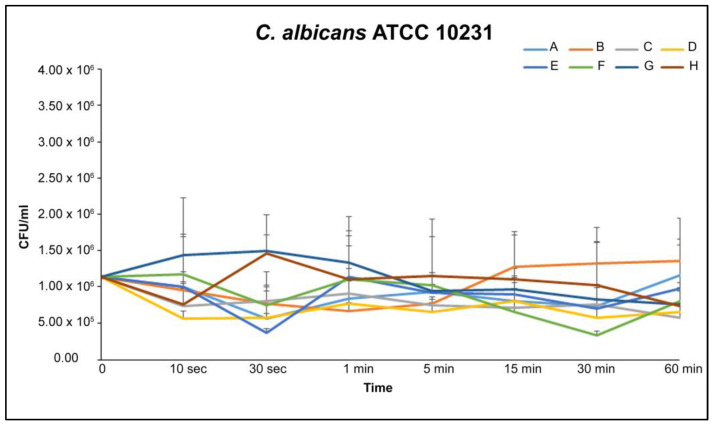
Time-kill curves with *C. albicans* ATCC 10231 for the A-H mouthwashes. For the legend of the name of mouthwashes see Table 1.

**Figure 8 antibiotics-12-00675-f008:**
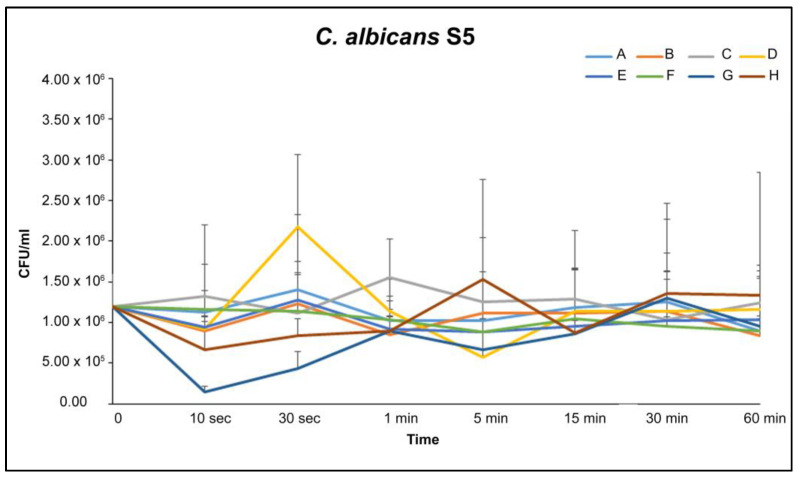
Time-kill curves with *C. albicans* S5 for the A-H mouthwashes. For the legend of the name of mouthwashes see Table 1.

**Figure 9 antibiotics-12-00675-f009:**
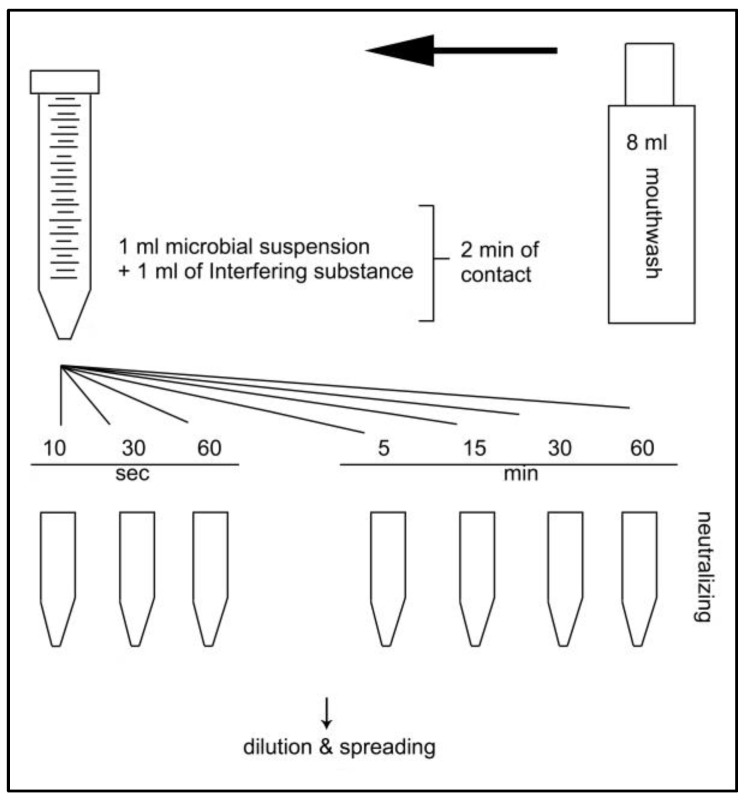
Representative image of Time Kill Curves.

**Table 1 antibiotics-12-00675-t001:** Mouthwashes used in the study.

Mouthwash	Composition	Manufacturer	LOTE
A—GUM^®^ PAROEX^®^	0.12% CHX + 0.05% CPC	GUM PAROEX, Saronno, Italy	I2112421
B—GUM^®^ PAROEX^®^	0.06% CHX + 0.05% CPC	GUM PAROEX, Saronno, Italy	T11
C—CURASEPT ADS 212	0.12% CHX + 0.05% CPC	CURASEPT S.p.A, Saronno, Italy	045 2
D—CURASEPT ADS 205	0.05% CHX	CURASEPT S.p.A, Saronno, Italy	041 2
E—PERIOAID^®^ Intensive Care	0.12% CHX + 0.05% CPC	DENTAID, S.L., Cerdanyola, Spain	S1142
F—PERIOAID^®^ Active Control	0.05% CHX + 0.05% CPC	DENTAID, S.L., Cerdanyola, Spain	S1033
G—Eludril Classic	0.10% CHX + 0.50% Chlorobutanol	Pierre Fabre ORAL CARE—Lisboa, Portugal	G00202
H—Eluday Care	0.05% CHX + 0.05% CPC	Pierre Fabre ORAL CARE—Boulogne, France	G00023

CHX = chlorhexidine gluconate; CPC = cetylpyridinium chloride.

**Table 2 antibiotics-12-00675-t002:** MIC and MBC/MFC values obtained with the tested mouthwashes (A-H) against *Enterococcus faecalis* ATCC 29212, *E. faecalis* SDL1, *Pseudomonas aeruginosa* ATCC 15442, *P. aeruginosa* BM, *Candida albicans* ATCC 10231, and *C. albicans* S5. Data were expressed as percentages. For the legend of the name of mouthwashes, see Table 1.

	MIC (%vol/vol)						MBC/MFC (%vol/vol)					
Mouthwashes	*E. faecalis*		*P. aeruginosa*		*C. albicans*		*E. faecalis*		*P. aeruginosa*		*C. albicans*	
	ATCC 29212	SDL1	ATCC 15442	BM	ATCC 10231	S5	ATCC 29213	SDL1	ATCC 15442	BM	ATCC 10231	S5
A	0.19	0.04	1.56	3.12	0.04	0.04	1.56	0.09	3.12	3.12	0.04	0.04
B	0.19	0.09	3.12	12.50	0.09	0.02	1.56	0.78	3.12	12.50	0.09	0.02
C	0.19	0.39	1.56	1.56	0.09	0.02	1.56	6.25	6.25	12.50	0.09	0.02
D	0.78	1.56	6.25	6.25	0.09	0.02	1.56	12.50	12.50	50	0.09	0.19
E	0.19	0.09	1.56	3.12	0.09	0.04	0.39	0.39	3.12	3.12	0.09	0.04
F	0.19	0.09	3.12	12.50	0.02	0.04	1.56	0.39	3.12	25	0.02	0.04
G	0.39	0.39	>50	>50	0.04	0.04	1.56	0.78	>50	>50	1.56	0.04
H	0.19	0.09	3.12	12.50	0.09	0.04	0.39	0.39	6.25	25	0.09	0.04

## Data Availability

Not applicable.

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
