# Peer review of "The Antibacterial and Antifungal Capacity of Eight Commercially Available Types of Mouthwash against Oral Microorganisms: An In Vitro Study"

_antibiotics, 2023, doi:10.3390/antibiotics12040675_

Round 1

Reviewer 1 Report

The manuscript "Antibacterial and Antifungal Capacity of eight Commercially Available types of mouthwash against oral microorganisms: an in vitro study" is a good and comprehensive work that evaluates the antibacterial activity against E. faecalis and P. aeruginosa and antifungal activity against C. albicans of 8 different commercial mouthwashes that contain chlorhexidine produced by Italian, Spanish and French manufacturers. It could be published in Antibiotics but firstly several issues need to be corrected:

1. For making the work more understandable and attractive for readers, it would be desirable to add a figure at the introduction with the molecular structure of chlorhexidine and cetylpyridinium chloride

2. The majority of the mouthwashes contain also cetylpyridinium chloride. It should also be mentioned in the introduction, in a similar way than chlorhexidine.

3. Please revise the abbreviations. For instance, CPC (cetylpyridinium) is not defined

4. The quality of all the figures is very poor. Also of Table 2. They need to be revise to increase their image quality and avoid pixelation.

5. End of the introduction. When the bacterial/fungal
strains are listed, Authors should specify which ones are standard cell lines, and which sensitive/resistant strains or isolates from clinical patients.

6. Page 5 - it has a big blank space, please avoid it to improve the quality of the presentation of the work.

7. Subsections can be added to the results/discussion to make it more readable. For example, "Mouthwashes evaluated", "Antibacterial activity", "Antifungal activity".

8. Mention briefly the manufacturers of the mouthwashes in the Materials and Methods section, for completion issues. As Table 1 is detailed, just something like "mouthwashes A, B, C, D were purchased from Y (city, country)" would we enough. And so on.

Author Response

The manuscript "Antibacterial and Antifungal Capacity of eight Commercially Available types of mouthwash against oral microorganisms: an in vitro study" is a good and comprehensive work that evaluates the antibacterial activity against E. faecalis and P. aeruginosa and antifungal activity against C. albicans of 8 different commercial mouthwashes that contain chlorhexidine produced by Italian, Spanish and French manufacturers. It could be published in Antibiotics but firstly several issues need to be corrected:

1. For making the work more understandable and attractive for readers, it would be desirable to add a figure at the introduction with the molecular structure of chlorhexidine and cetylpyridinium chloride

AUTHORS’ANSWER: Thank you very much for your comment. As you suggested, we added the molecular structure of chlorhexidine and cetylpyridinium chloride (Figure 1 and 2) in the Introduction section.

  1. The majority of the mouthwashes contain also cetylpyridinium chloride. It should also be mentioned in the introduction, in a similar way than chlorhexidine.

AUTHORS’ANSWER: Thank you very much for your comment. As you suggested, we added the following sentences in Introduction section regarding the CPC:

“Cetylpyridinium chloride (1-hexadecylpyridinium chloride) (CPC) (Figure 2), a quaternary ammonium compound, is also frequently used in dental practice in different mouthwashes and dentifrices. Its antimicrobial action is correlated to hydrophobicity of the side chain and with its capability to damage the cell membrane affecting the growth of Gram-positive and Gram-negative bacteria. It also showed a remarkable anti-biofilm action against of methicillin resistant Staphylococcus aureus, Streptococcus mutans and Veillonella parvula biofilm formation”

  1. Please revise the abbreviations. For instance, CPC (cetylpyridinium) is not defined

AUTHORS’ANSWER: Thank you very much for your comment. As you suggested, in the new version of the MS, we better specified the abbreviations in the Introduction section and in the Table 1.

  1. The quality of all the figures is very poor. Also of Table 2. They need to be revise to increase their image quality and avoid pixelation.

AUTHORS’ANSWER: Thank you very much for your comment. As you suggested, we improved the quality of all figures and the table.

  1. End of the introduction. When the bacterial/fungal strains are listed, Authors should specify which ones are standard cell lines, and which sensitive/resistant strains or isolates from clinical patients.

AUTHORS’ANSWER: Thank you for the comment. As you suggested, we inserted a sentence at the end of introduction specifying the standard cell lines and the resistance profiles of the clinical strains in the M&M section.

In Introduction section “The references strains used in this work are the main microorganisms test used for the evaluation of bactericidal, fungicidal of antiseptic medicinal products (European pharmacopeia 10.0 07/2015:50111).”

In Materials and methods section “The clinical strains were characterized for their antimicrobial susceptibility.  C. albicansS5 was sensible to antifungal drugs commonly used in therapy. E. faecalis SDL1 was resistant strain to Amoxicillin, Vancomycin. P. aeruginosa BM was resistant strain to Amikacin, Amoxicillin, Aztreonam, Cefotaxime, Ceftazidime, Ceftriaxone, Cefalotin, Ciprofloxacin, Gentamicin, Levofloxacin, Nitrofurantoin, Piperacillin, Tobramicin.”

  1. Page 5 - it has a big blank space, please avoid it to improve the quality of the presentation of the work.

AUTHORS’ANSWER: Thank you very much for your comment. As you suggested, in the new version of the MS, we deleted it.

  1. Subsections can be added to the results/discussion to make it more readable. For example, "Mouthwashes evaluated", "Antibacterial activity", "Antifungal activity".

AUTHORS’ANSWER: Thank you very much for your comment. As you suggested, in the new version of the MS, in the Results section, we inserted the subsections.

  1. Mention briefly the manufacturers of the mouthwashes in the Materials and Methods section, for completion issues. As Table 1 is detailed, just something like "mouthwashes A, B, C, D were purchased from Y (city, country)" would we enough. And so on.

AUTHORS’ANSWER: Thank you very much for your comment. As you suggested, we inserted the following sentence in the Materials and Methods section.

“The mouthwashes A and B were purchased from GUM PAROEX (Saronno, Italy); C and D from CURASEPT S.p.A (Saronno, Italy); E and F from DENTAID, S.L. (Cerdanyola, Spain); G and H from Pierre Fabre ORAL CARE (Lisboa, Portugal).”

Reviewer 2 Report

Dear Authors,

The work presented in your manuscript is not original and the novelty is totally missing. 

The antimicrobial effect of chlorhexidine from mouthwashes is well known, since the introduction of this antiseptic in therapy. There is nothing new here.

Plus, the use of cetylpyridinium chloride results in a synergistic activity, improving the antimicrobial potential. Nothing new here too.

Candida albicans is not a bacteria, as you mention in line 77!

Table 2 is not clear at all.

The antimicrobial activity includes the antifungal one (correct in line 322).

The Conclusion part has to be detailed.

Author Response

The work presented in your manuscript is not original and the novelty is totally missing. 

The antimicrobial effect of chlorhexidine from mouthwashes is well known, since the introduction of this antiseptic in therapy. There is nothing new here.

Plus, the use of cetylpyridinium chloride results in a synergistic activity, improving the antimicrobial potential. Nothing new here too.

AUTHOR’S ANSWER: Dear Revisor, I am very sorry that you have not understood the aim of this study.  In the oral cavity, one of the main causes of failure of traditional treatments is the long-term use of antiseptics which can produce antimicrobial resistance in vivo due to exposure to sublethal concentrations which has occurred in the last century. Antimicrobial resistance is a global challenge and it is important to control and prevent infection associated with resistant microorganisms. For this reason, in our work we evaluated and compared the antimicrobial action and the effects over time of 8 types of mouthwash against reference and clinical strains of the main resistant microorganisms responsible for oral cavity pathologies.

We have modified the text of the manuscript in order better to specify this problem.

In the Introduction section:

“A long-term use of antiseptics may produce resistance in vivo due to the exposure to sublethal concentrations that has arisen over the last century [22]. The antimicrobial resistance is a global challenge and in particular, an increasing in antimicrobial resistant microorganisms in the oral cavity is the main cause of the failure of the traditional treatments. The oral cavity is a ready environment for horizontal gene transfer because of the close proximity of bacteria in plaque and the availability of exogenous DNA passing through the oral cavity [23]. For this reason, it is important to control and prevent the infection associated to resistant microorganisms.”

“This work aimed to evaluate and compare the antimicrobial action and the effects over time of 8 types of mouthwash against reference and clinical strains of the main resistant microorganisms responsible for oral diseases: Enterococcus faecalis ATCC 29213, E. faecalis SDL1, Pseudomonas aeruginosa ATCC 15442, P. aeruginosa BM, Candida albicans ATCC 10231 and C. albicans S5. The references strains used in this work are the main microorganisms test used for the evaluation of bactericidal, fungicidal of antiseptic medicinal products (Euopean pharmacopeia 10.0 07/2015:50111). The final aim was to suggest mouthwashes able to prevent and control the microbial proliferation tackling the antimicrobial resistance.”

In the Abstract

“The results demonstrate significant differences in the antimicrobial action of the tested mouthwashes, although all contained chlorhexidine and most of them also cetylpyridinium chloride. The relevant antimicrobial effect of all tested mouthwashes with the best higher antimicrobial action was recorded by A - GUM® PAROEX®A and B - GUM® PAROEX® considering their effect against resistant microorganisms and their MIC values.

In the Conclusions

“The results demonstrate significant differences in the antimicrobial action of the tested mouthwashes, although all contained chlorhexidine and most of them also cetylpyri-dinium chloride. The higher antimicrobial action was recorded by A - GUM® PARO-EX®A and B - GUM® PAROEX® considering their effect against resistant microorgan-isms and MIC values.

The obtained results suggest that although the presence of CHX and CPC in different concentrations provides effective antimicrobial action, additional molecules in the mouthwash composition can significantly empower or diminish their effectiveness.

Regular and correct oral hygiene procedures with the addition of suitable mouthwashes in terms of concentrations and efficacy against resistant microorganisms, can be considered a real solution to control and prevent infections of the oral cavity.

Future work will be done to confirm these results with other oral aerobic and anaerobic microorganisms associated with oral diseases”.

Candida albicans is not a bacteria, as you mention in line 77!

AUTHOR ANSWER: Thank you very much for your comment, the sentence has been corrected in:

“In addition to several other studies in which chlorhexidine gluconate was shown to be effective in reducing the virulence of bacteria and yeasts such as E. faecalis and C. albicans”.

Table 2 is not clear at all.

AUTHOR ANSWER: Thank you for your comment. We modified the Table 2 regarding the values of MIC and MBC/MFC (%vol/vol) of the compared mouthwashes, against the different microorganisms.

The antimicrobial activity includes the antifungal one (correct in line 322).

AUTHOR ANSWER: Thank you very much for your comment. The word “antifungal “ has been removed.

The Conclusion part has to be detailed.

AUTHOR ANSWER: Thank you very much for your comment; as shown in previous points, the conclusions section has been modified.

Round 2

Reviewer 1 Report

The manuscript has improved significantly and the concerns have been adequately addressed.

Just as a minor comment - that can be addressed in article processing, Table 2 has very small font size. To increase its readability, the figure can be enlarged to occupy all page width. MDPI journals allow the use of the left margin in these casese to improve the quality of presentation and the readability of the work.

Author Response

The manuscript has improved significantly and the concerns have been adequately addressed.

Just as a minor comment - that can be addressed in article processing, Table 2 has very small font size. To increase its readability, the figure can be enlarged to occupy all page width. MDPI journals allow the use of the left margin in these casese to improve the quality of presentation and the readability of the work.

AUTHORS’ANSWER: Thank you very much for your comment. As you suggested, we increased the size of the table 1 and improved its quality.

Reviewer 2 Report

Dear Authors,

I understood very well what you wrote in the manuscript, that is why I made all the corrections and suggestions in my previously report.

You say: "This work aimed to evaluate and compare the antimicrobial action and the effects over time of 8 types of mouthwash based on chlorhexidine..." How do you quantify the effects over time? Your test is done until 60 minutes. When you say that you follow what happens with something over time that means you take subjects that are using chlorhexidine mouthwash over years and you check what is changed from time to time, for example from one year to another.

Sorry, but your work, in my opinion, is lacked of clarity and novelty. I suggest that you change your work protocol and really follow what happens after the use for a long period of time with this type of products.

Author Response

Dear Authors,

I understood very well what you wrote in the manuscript, that is why I made all the corrections and suggestions in my previously report.

You say: "This work aimed to evaluate and compare the antimicrobial action and the effects over time of 8 types of mouthwash based on chlorhexidine..." How do you quantify the effects over time? Your test is done until 60 minutes. When you say that you follow what happens with something over time that means you take subjects that are using chlorhexidine mouthwash over years and you check what is changed from time to time, for example from one year to another.

Sorry, but your work, in my opinion, is lacked of clarity and novelty. I suggest that you change your work protocol and really follow what happens after the use for a long period of time with this type of products.

AUTHORS’ANSWER: Thank you very much for your comment. We appreciated your comment and we modified the MS according your suggestion. We agree with you that a long-term in-vivo study about the use of chlorhexidine could provide interesting information, and this could be a viable option for another future work. However, in this case, we have conducted an in-vitro study, in which we have compared different products, by using microbial analysis and respecting the times that are described in previous literature. The time-kill curve is conducted at different time and this is a standard microbial test validated by literature, the novelty of this study it is in the comparison among these products that to our knowledge has never been tested. Another novelty is in the results, because usually patients and dentists could believe that mouthwashes with similar percentages of chlorhexidine and CPC could have the same effects on bacteria, but in this study we have shown that there are significant differences among them.   Basic research is based on the use of validated tests and protocols that are permitted to compare novel products.

To avoid misunderstanding, we modified the aims underling that we evaluated the antimicrobial effect at different contact times (10 sec, 30 sec, 60 sec, 5 min, 15min, 30min, 60min) and not over time.

Round 3

Reviewer 2 Report

Dear Authors,

Thank you for considering my comments and suggestions and for making the changes on your manuscript.

I suggest that you continue the investigation of this products and add an in vivo assay, too. So, you will have a closer look on the effects of Chlorhexidine mouthwashes on patients and also evaluate them over time.

Author Response

Dear Reviewer and Editor,

We would like to sincerely thank the reviewers for his/her detailed comments, which were very useful for improving our manuscript. We have taken all these comments into account in the revised version of the paper. The following are the replies to reviewers’ comment. In the answers, the reviewer's comments are in black and our answers are in bold.

I hope the answers will satisfy the reviewer. I think the in vivo study will be very important to understanding the effects of mouthwashes on the oral microbiota, but as indicated in the title, this manuscriot is an in vitro study. In the new version of the MS, I inserted this work limitation.

Best regards.

reviewer 2

Thank you for considering my comments and suggestions and for making the changes on your manuscript.

I suggest that you continue the investigation of this products and add an in vivo assay, too. So, you will have a closer look on the effects of Chlorhexidine mouthwashes on patients and also evaluate them over time.

AUTHORS’ANSWER: Thank you very much for your comment. We appreciated your comment and we will perform an in vivo study in the next work.  In fact, we have already initiated the request to the "Ethics Committee" to approve the in vivo study and we are awaiting an answer to enroll the patients. Since we believe correct your suggestion, in the new version of the MS, we inserted the work limitation relating to the lack of the in vivo study, underling the correspondence with the title that is an in vitro study.